# Surface Acoustic Wave Humidity Sensor: A Review

**DOI:** 10.3390/mi14050945

**Published:** 2023-04-27

**Authors:** Maria Muzamil Memon, Qiong Liu, Ali Manthar, Tao Wang, Wanli Zhang

**Affiliations:** 1State Key Laboratory of Electronic Thin Films and Integrated Devices, School of Integrated Circuit Science and Engineering, University of Electronic Science and Technology of China (UESTC), Chengdu 610054, China; 2School of Mechanical and Electrical Engineering, University of Electronic Science and Technology of China (UESTC), Chengdu 610054, China

**Keywords:** SAW, humidity sensor, humidity sensing materials, performance characteristics

## Abstract

The Growing demands for humidity detection in commercial and industrial applications led to the rapid development of humidity sensors based on different techniques. Surface acoustic wave (SAW) technology is one of these methods that has been found to provide a powerful platform for humidity sensing owing to its intrinsic features, including small size, high sensitivity, and simple operational mechanism. Similar to other techniques, the principle of humidity sensing in SAW devices is also realized by an overlaid sensitive film, which serves as the core element whose interaction with water molecules is responsible for overall performance. Therefore, most researchers are focused on exploring different sensing materials to achieve optimum performance characteristics. This article reviews sensing materials used to develop SAW humidity sensors and their responses based on theoretical aspects and experimental outcomes. Herein the influence of overlaid sensing film on the performance parameters of the SAW device, such as quality factor, signal amplitude, insertion loss, etc., is also highlighted. Lastly, a recommendation to minimize the significant change in device characteristics is presented, which we believe will be a good step for the future development of SAW humidity sensors.

## 1. Introduction

Humidity is an important environmental variable that plays a decisive role in our daily lives. In addition to human comfort, its detection and control are necessary for various manufacturing and storage environments to achieve optimum quality in production [1,2,3,4]. For instance, in the textile industry, the environment should be kept in damp conditions to prevent fabric material from clinging. For electronic manufacturing processes, a clean room requires dry conditions. In pharmacist industries, adequate humidity conditions are essential for packaging, storage, and transport. These sensors play a vital role in soil moisture detection, greenhouse humidity monitoring, dew prevention, storage, etc., in the agricultural industry. They are also used in medical fields in respiration monitoring, infusion pumps, ventilators, diagnostic instruments, medicine storage, etc. Furthermore, their utilization is significant in electronic manufacturing, automotive, food quality monitoring, and chemical and material processing industries. According to the global market report, the humidity sensor market reached $4.7 billion in 2020 and is expected to expand by nearly $10.4 billion in 2026.

To fulfill humidity detection demands, great efforts have been made to develop humidity sensors based on different operational mechanisms. According to the sensing techniques, these sensors are typically categorized as capacitive [5,6], resistive [7,8], optical [9,10], and acoustic [11,12,13,14]. The classification of humidity sensors based on sensing mechanisms is shown in Figure 1. Table 1 summarizes the basic mechanism and advantages and disadvantages of different types of humidity sensors.

For practical implementation, the humidity sensor must possess satisfactory traits, including high sensitivity, wide detection range, low hysteresis, good linearity, good repeatability, quick response, and short recovery time. Additionally, the factors such as compact size, low power consumption, low cost, and compatibility with fabrication processes pave the way to make the sensor design an optimal choice for practical applications. Acoustic sensors have already been found to possess these inherent characteristics.

Among different types of acoustic wave devices, Surface acoustic wave (SAW) devices have captured wide attention owing to their simple detection mechanism, easy integration, high sensitivity, good stability, small size, low cost, high accuracy, planar structure, and capability to operate in wireless mode [19,20,21]. These features make the SAW devices appropriate for detecting various physical parameters such as pressure, temperature, gas, vapors, humidity, and motion [22,23,24,25,26]. The principle of humidity sensing is realized by covering the SAW device configured as a delay line or resonator with a sensing film, as shown in Figure 2. Both of these configurations possess similar output characteristics and the same operational mechanism.

The overlaid sensing film is the main determining factor to attain maximum effectiveness of the SAW humidity sensor. Therefore the researchers are mainly concentrated on exploring different sensing materials to optimize the parameter of interest. This article reviews the SAW humidity sensors developed in nearly three decades. Firstly the basic mechanism is elaborated, followed by a detailed review of humidity-sensitive materials. Herein, data on device design, sensing materials, and performance characteristics of various existing SAW humidity sensors are summarized in tables. Moreover, the influence of overlaid sensing materials on the characteristics of SAW devices is addressed in this article. Lastly, a recommendation is presented for the future development of SAW humidity sensors.

## 2. Sensing Mechanism

The operation of humidity detection is realized by a sensing film overlaid on the surface of the SAW device. Any perturbation on the surface due to the absorption/desorption of water molecules affects the intrinsic parameters (mass density, thickness, conductivity, and elasticity) of the sensing film, which alters the wave characteristics. Figure 3 illustrates the operational mechanism of humidity sensing, where the delay line configured SAW device is coated with a sensing layer over the delay path, as shown in Figure 3a. With the application of an electrical signal to input IDTs, a surface wave is generated and propagated at a specific velocity towards the output IDTs, as shown in Figure 3b. The overlaid sensing materials possess sensitive signal to water molecules, so the absorption/desorption of these molecules influence their properties and cause perturbation on the surface, altering the acoustic wave velocity and attenuation as illustrated in Figure 3c,d. This perturbation in wave characteristics for different humidity levels can be detected by measuring frequency or phase shift at the output IDTs, where the propagating wave is converted back to an electrical signal, as depicted in Figure 3e.

The variation in propagating acoustic wave velocity (v) due to the interaction between sensing film and vapor molecules occurs by three operational mechanisms, i.e., mass loading, acoustoelectric and elastic loading, which can be formulated as [27,28]
(1)$$\frac{∆v}{v_{0}}={-c}_{m}f_{0}∆\left(\frac{m}{A}\right)+4c_{e}\frac{f_{0}}{v_{0}^{2}}∆\left(hG^{\prime }\right)-\frac{K^{2}}{2}∆\left[\frac{\sigma_{s}^{2}}{\sigma_{s}^{2}+v_{0}^{2}C_{s}^{2}}\right]$$

Here, $c_{m}$ and $c_{e}$ are mass and elasticity sensitivity coefficients of the substrate, respectively. $f_{0}$ and $v_{0}$ are the center frequency and unperturbed velocity, respectively; m/A is the mass change per unit area, *h* is the sensing film thickness, $G^{\prime }$ is the storage modulus, $K^{2}$ is the electromechanical coupling coefficient, $\sigma_{s}$ is the sheet conductivity of the sensing film and ${\;C}_{s}$ is the capacitance per unit length of the piezoelectric substrate.

In Equation (1), the first term represents the variation in propagating wave velocity caused by the mass loading effect, i.e., mass density change due to the interaction of vapor molecules within the sensing film. The second term characterizes the alteration in wave velocity due to elastic loading, i.e., the modulus change. The last term illustrates the variation in wave velocity by the acoustoelectric effect, the change in electrical properties such as conductivity and permittivity. All these changes in velocity can be monitored in terms of frequency or phase shift. The algebraic signs in Equation (1) illustrate that the velocity shifts positively with elastic loading and negatively with mass and electric loading. The effect of elastic loading on the sensor response is mostly neglected for semiconducting metal oxides, ceramics, nanostructured and carbon-based materials, as the alteration in mechanical properties due to the vapor molecules absorption is negligible for these sensing materials. Therefore, the mass loading effect is mainly considered the dominant mechanism for SAW humidity sensors. However, in earlier studies, some researchers had observed a significant effect of elastic loading, particularly for viscoelastic polymer films [19,29]. Furthermore, wave velocity had observed to decrease in some regimes while found to increase in other regimes. The contribution of the elastic loading effect cannot be completely ignored for polymer sensing films, as the absorption of vapors can greatly influence their mechanical properties.

In most of the polymer-based SAW humidity sensors, the elastic effect contributes in the same direction as the mass loading effect, i.e., the decrease in velocity (negative frequency shift), mainly due to the reduction in the modulus [30,31,32], known as plasticization or softening effect. In that case, the water molecules act as a spacer between polymer chains. However, for some specific polymers, the absorption of water molecules hardens the polymer chain instead of the usual softening effect, known as antiplasticization or stiffening effect. This unusual hardening effect increases the velocity that causes the elastic effect contributes in the opposite direction to the mass loading effect, as observed for the fluorinated polyimide-based SAW humidity sensor [33]. Likewise, the positive shift was observed for the polyimide-coated SAW humidity sensor initially in lower ranges which was overwhelmed by the mass loading effect [34]. Generally, the type of sensing material and its interaction with water molecules characterize the dominant mechanism and the total response of the sensor. The next section covers a detailed review of the performance characteristics and responses of SAW humidity sensors based on the different sensing materials.

## 3. Sensing Materials

The sensing material coated over the surface of the SAW device is an integral part of humidity detection, which has the sensitivity to grab water molecules and give rise to sensor response based on operational mechanisms (Mass loading, acoustoelectric and elastic loading). For implementation, sensing material is required to meet a variety of specifications such as strong adhesion with the substrate, good ability to absorb water vapors, long-term repeatability, lengthy operational lifetime, repeatable, quick, and reversible interaction with the water molecules [35,36]. Therefore, the interest of researchers is specifically in exploring and altering humidity sensing materials to get optimal sensing properties. The humidity-sensing materials are mainly categorized into metal oxides [37,38], carbon-based [13,39], ceramics [38,40], nanostructured [41,42], and polymers [43,44,45]. Figure 4 illustrates the classification of sensing materials commonly utilized for SAW humidity sensors.

### 3.1. Polymers

Polymers have become one of the most extensively used humidity-sensitive materials for over 30 years. They have gained particular attention as humidity sensing layers for SAW devices owing to their high response to water molecules, room temperature operation, low cost, and simple coating techniques. Up to present, polyimide (PI), poly(vinylalcohol) (PVA), polyXIO, poly(N-vinylpyrrolidone)(PVNP), polyaniline (PANI), conjugated polymers, poly(p-diethynylbenzene) (PDEB), sodium polysulfonesulfonate (NaSPF), polyvinyl pyrrolidone (PVP) and sulfonated tetrafluorethylene copolymer(Nafion) are reported as humidity sensing membranes in SAW devices [30,31,43,45,47,48,49,50,51]. The major focus of the researchers is to characterize the sensing performance of polymers deposited using different techniques on different SAW structures operating at different frequencies, as listed in Table 2. Generally, methods such as spin coating, dip coating, solvent casting, electrospray, vacuum evaporation, etc., have been exploited for the deposition of polymers.

Li et al. [50] investigated the sensitivity of poly(p-diethynylbenzene) (PDEB) and sodium polysulfonesulfonate (NaSPF) films in 20–85% relative humidity range at different temperatures. PDEB and NaSPF were deposited on the 128° YX–LiNbO3-based delay line SAW device with Langmuir–Blodgett (LB) and spin coating techniques, respectively. The sensitivity of NaSPF-based SAW sensors was found to increase with temperature, i.e., −0.9 and −1.1 kHz/%RH at 22 °C and 30 °C, respectively. The sensitivity of PDEB-based SAW sensors was obtained to decrease with temperature, i.e., −0.4 and −0.36 kHz/%RH at 22 °C and 30 °C, respectively. Moreover, NaSPF based SAW humidity sensor showed higher sensitivity due to its hydrophilic and hygroscopic nature. On the contrary, PDEB is hydrophobic, exhibiting weak affinity towards water molecules, resulting in low sensitivity. Their finding suggests that the absorption ability of the sensing film is an important factor to be considered for obtaining a high response. Buvailo et al. [31] reported polyvinyl pyrrolidone (PVP) thin film-based SAW humidity sensors with the fastest response. The response and recovery times from 5 to 95%RH step were calculated as 1.5 s and 2.5 s, respectively. Moreover, they described mass loading and viscoelastic effects as the main mechanisms for the response. However, in [52], the change in conductivity (acoustoelectric effect) was reported as a dominant mechanism rather than mass or viscoelastic for polystyrene sulfonic acid sodium film, which is a hygroscopic polymer. Similarly, the change in permittivity was characterized as the main mechanism for cellulose acetate based humidity sensor [53]. However, some authors described the elastic effect as the main contributor [30,33]. In general, the overall response of the sensor is primarily defined by operational mechanisms, which depend on the physical and chemical properties of the sensing film and the influence of water vapor interaction on their intrinsic parameters such as mass density, thickness, conductivity, and elasticity.

In addition to exploring polymers as humidity-sensitive materials, some researchers have presented the possibilities of improving the performance characteristics by altering the physical parameters of the sensing layer, such as thickness, densities, and mechanical adhesion [32,54,55]. In [31], the humidity tests for thinner and thicker PVP and PVA films coated SAW devices were investigated. The thicker films coated sensors showed larger responses with good resolution, illustrating that the thickness of the film is a key factor in obtaining good resolution and large response. Ndao et al. [56] optimized the sensing range of PVA-coated sensors with different mass percentages, i.e., 2.5%, 5%, and 10. The sensor coated with 10% PVA film presented a high response due to more hydroxyl groups but was only sensitive up to 95%. On the contrary, 2.5% PVA film was sensitive in the higher range of up to 98% but insensitive below 60%. The 5% PVA sensing film was observed to detect humidity in the wide range of 10–98%. Moreover, they observed better measurement precision and lower hysteresis with the increase in the density of PVA film. Similarly, Alam et al. [55] attempted to improve the dynamic range by varying film thickness using different PVA solutions (5%, 2%, and 1% solution). They showed that a wide dynamic range could be obtained with small mass loading by decreasing the thickness of the PVA film. The sensitivity was found to increase with increasing thickness of PVA, the same as [56]. Contrarily, they observed increased hysteresis for thicker PVA film, possibly due to the different densities or the different coverage areas of the sensing film. Generally, thinner PVA films provide a wide sensing range but low sensitivity compared to thicker films. However, the effect of film thickness on sensing characteristics is not identical for all polymers. In [32], the sensitivity to humidity was found to increase with decreasing the thickness of the Nafion polymer. The Nafion films with 50 nm, 100 nm, and 150 nm thicknesses were spin-coated on quartz-based SAW resonators. They observed the highest sensitivity with the film thickness of 50 nm and recommended the thickness should not exceed 100 nm for Nafion.

Apart from physical parameters, the modification in the chemical properties of the sensing film can also aid in improving the sensing characteristics. In [33], the polyimide with hydrophobic characteristics was used as the humidity-sensing film to reduce the hysteresis issue. The fluorinated polyimide-coated humidity sensor exhibited negligible hysteresis compared to ordinary polyimide-coated sensors [43,57]. Their findings suggested that the chemical modifications by introducing hydrophobic functional groups can help to reduce the hysteresis issues that are more pronounced in hydrophilic or hygroscopic polymers due to the strong bonding between water molecules and sensing film. The hydrophilic polymers possess a high absorbing ability but may result in poor stability at high humidity levels due to the dissolution of the sensing film.

**Table 2 micromachines-14-00945-t002:** Polymer-based SAW humidity sensors.

Configuration	Substrate	Frequency (MHz)	Polymer	Deposition Technique	Operational Mechanism	Ref
Delay line	128° YX liNbO_3_	30	Polystyrene sulfonate acid sodium	-	Acoustoelectric	[52]
Delay line	Quartz	80	PI	Spin coating	Mass loading	[43]
Delay line	YZ-cut LiNbO_3_	50	PolyXIO	Drop coating	Mass and Viscoelastic loading	[30]
Delay line	ZnO/SiO_2_/Si	-	Platinum polyyne (Pt-P-HDOB)	Spin coating	-	[47]
Delay line	YZ-cut LiNbO_3_	250	PVP	Spin coating	Mass and Viscoelastic loading	[31]
Two port resonator	ST-X Quartz	300	PANI	Spin coating	Viscoelastic and electric loading	[45]
Two port resonator	Quartz	194	Nafion	Electrospray and thermal evaporation	-	[58]
Two port resonator	ST-X cut Quartz	433	Polyelectrolyte APTS-P	Electrospray	Mass, electric, and viscoelastic	[59]
One port resonator	ST Quartz	434	Nafion	Spin coating	Mass and Viscoelastic loading	[32]
One port resonator	AlN/SOI	153	Fluorinated PI	Spin coating	Mass and stiffening effect	[33]
Delay line	128° YX liNbO_3_	138	PDEP	Langmuir-Blodgett (LB)	Mass loading	[50]
Delay line	128° YX liNbO_3_	138	NaSPF	Spin coating	Mass and electric loading	[50]
One port resonator	-	433.92	PVA	Dip coating	-	[55]
Delay line	128° YX liNbO_3_	30	PVA	Dip coating	Mass, electric, and viscoelastic	[56]

Furthermore, selecting the proper coating technique is another essential factor for optimizing performance characteristics. JASEK et al. [58] characterized the humidity sensing properties of Nafion^®^ film deposited with two different techniques, i.e., thermal evaporation and electrospraying. The film deposited by the vacuum evaporation method agglomerated into bubbles, while the electrospray-deposited film was uniform. Moreover, the electrospray Nafion^®^ film showed lower sensitivity and detection limit but quick response time compared to thermal evaporated Nafion^®^ film. Apart from deposition techniques, the conditions and parameters of the deposition methods also affect the humidity response of the sensor. Li et al. [59] examined the effect of electrospray parameters on the response of polyelectrolyte-coated SAW humidity sensors. They observed low humidity sensitivities with short (10 min) and long (30 min) electrospray times. Similarly, low sensitivities were obtained with low (5 mg/mL) and high (50 mg/mL) electrospray solution concentrations. The sensor with 20 min electrospray time and 25 mg/mL electrospray solution resulted in higher sensitivity, as shown in Figure 5. Similarly, for the spin coating technique, different spinning rates result in different formations of films, which affect the response of the sensor as examined for PVA and PVP films [31]. The PVP films formed at higher spinning rates resulted in thinner coatings and appeared more sensitive in low RH levels but less responsive at higher RH ranges. However, the lower spinning rates resulted in thicker films which exhibited higher sensitivity but a slower response and recovery time. The high sensitivity of thicker films is associated with the capacity to absorb large amounts of water molecules, which also require a long time to release, resulting in slower response and recovery time. These results suggest that the deposition conditions and parameters could greatly affect the response of the sensor.

To gain the maximum effectiveness of the humidity sensor, it is of utmost importance to carefully select various factors such as thickness, density, physical and chemical properties of sensing film, deposition technique, and their condition and parameters.

### 3.2. Metal Oxides and Ceramics

Polymer sensing films show good sensing characteristics at room temperature, but their stability and repeatability are adversely affected in different environmental conditions, especially at higher temperatures. To overcome stability issues, metal oxides are utilized as an alternative due to their high thermal stability. Metal oxides such as ZnO, Co_3_O_4_, CuO, TiO_2,_ and SnO_2_ have been explored as the humidity sensing films for SAW devices. Among these, ZnO is versatile and one of the most extensively investigated metal oxides. The nanocrystalline ZnO [37,38,60], Ga-doped ZnO [61], and different nanostructures of ZnO [62,63] are explored for humidity detection. Hong et al. [60] examined the humidity sensing of ZnO films annealed at 500 °C and 600 °C and observed the largest frequency shift for the films annealed at 500 °C due to the decrease in grain size, which results in a large surface area for water molecules adsorption. The increase in annealed temperature resulted in large particle and pore size of ZnO thin films, which affects the sensitivity due to alteration in surface area for water molecules absorption. Moreover, they observed an increased frequency shift from 120 to 170 kHz by increasing the thickness of ZnO film from 180 to 300 nm. The thicker and more porous films are advantageous for obtaining large responses. As well as metal oxides, ceramics such as silica (SiO_2_), Al_2_O_3_, etc., are also used as humidity-sensing membranes due to their porous nature, which is advantageous to provide a large surface area for water molecules absorption. Tang et al. [38] compared humidity detection between metal oxides (ZnO, Co_3_O_4_, CuO, TiO_2_) and SiO_2_ films coated on quartz SAW resonators. They obtained much higher responses with sol-gel SiO_2_ films and observed the increased response with the increase in SiO_2_ film thicknesses, as shown in Figure 6a,b. The high sensitivity was associated with the porous structure of SiO_2_ film, which facilitates the absorption of water molecules due to the large surface area. On the contrary, the metal oxide films were smooth, dense, and less porous, as shown in SEM and AFM images in Figure 7a,b. Alumina (Al_2_O_3_) is another porous ceramic material utilized as the humidity-sensing membrane for SAW devices [64,65].

### 3.3. Nanostructured and Carbon Based

In recent years, the improved sensitivity based on large surface area escalated the development of nanostructured coatings as the humidity sensing material for SAW sensors. Till now, a variety of nanostructures has been synthesized, including nanofibers [41,66,67], nanorods [62,63], nanobelt [68], nanoclusters [69], nanowires [70], nanoparticles [42,68], oxide and composites nanomaterials [71,72,73,74]. Lin et al. [66] presented a highly sensitive and ultrafast response sensor coated with Electrospun PANI/PVB composite nanofibers. They obtained a sensitivity of 75 kHz/%RH in the 20–90%RH range with response and recovery time of 1 s and 2 s, respectively. Sheng et al. [67] reported one of the most highly sensitive humidity sensors with a multi-walled carbon nanotube/Nafion (MWCNT/Nafion) composite as a sensing material. The high sensitivity of 427.6 kHz/%RH in the 10–80% RH range had achieved due to the three-dimensional (3D) porous structure. Compared to conventional bulk materials, the nanostructured sensing films provide much large surface areas, which offer more active sites for the anchoring of water molecules, leading to large diffusion depth, thereby improving the sensing characteristics. Moreover, synthesizing nanostructures at different conditions can greatly enhance the sensing characteristics. Li et al. [42] presented that the chemically deposited Ag films exhibit high sensitivity due to their porous and rough structure, while the films deposited by thermal evaporation were smooth and not noticeably responsive to humidity. They obtained improved sensitivity with the formation of Ag nanoparticles by chemically synthesized and post-synthesized annealed films at 400 °C for 1 hr. Moreover, the sensitivity of post-synthesized annealed films was enhanced by 4 times (from 207 to 850 kHz) when the reaction time was increased from 2 min to 6 min due to an increase in particle size from 260 to 330 nm. The nanoparticle size and film morphology are the key factors in obtaining high sensitivity.

Likewise, owing to the large surface-to-volume ratio and the presence of hydrophilic functional groups, graphene oxide (GO) has become fascinating material for developing humidity sensors. Graphene oxide is a carbon-based material with oxygen-containing, i.e., carboxyl groups, which are hydrophilic functional groups that provide the feasibility of water molecules absorption. Balashov et al. [75] compared the humidity sensitivity of PVA and GO-coated SAW devices. The sensitivity of thin PVA film was obtained as 0.47 kHz/%RH and 1.54 kHz/%RH for GO film, which is almost three times larger than PVA coated sensor. The mass loading effect was described as the main contributor to the sensor response. Le et al. [13] achieved a sensitivity of 25.3 kHz/%RH with uniform and highly oxidized GO film transferred to the SAW device based on AlN/Si layered structure. Their findings demonstrate that the uniformity of sensing materials should be optimized to achieve high sensing performance. Su et al. [39] presented a layered structure of humidity sensing films consisting of a three-dimensional architecture of graphene/polyvinyl alcohol/silica (3DAG/PVA/SiO_2_) and applied it in respiration monitoring. High sensitivity was obtained by the synergetic effect of large diffusion/viscoelastic/mass loading effects with the arrangement of 3DAG/PVA/SiO_2_ layered structure, respectively.

Graphene-based sensing materials provide good sensitivity, but surface contamination, long-term drift, poor reproducibility, environmental pollution, and explosion hazards are some of their big disadvantages. Each kind of sensing material has advantages and disadvantages. Table 3 summarizes the performance characteristics of some existing SAW humidity sensors based on different sensing materials.

## 4. Practical Application Cases

The main traits of humidity sensors are high sensitivity, low hysteresis, fast response time, quick recovery time, stability, durability, linearity, and wide sensing range. However, the conditions for humidity detection depend on the specific application. Breath monitoring and skin humidity detection are among the main applications of humidity sensors for human healthcare detection. Some researchers have successfully implemented SAW humidity sensors for respiration monitoring and noncontact detection of finger motion [39,69,73,78]. Rimeika et al. [69] detected the moisture content in the exhaled air for human respiration monitoring by utilizing bovine serum albumin (BSA) and bovine serum albumin-gold nanoclusters (BSA-Au NCs) films deposited on LiNbO_3_ SAW delay line. Similarly, Su et al. [39] developed a SAW humidity sensor based on the layered sensing film structure of three-dimensional architecture graphene/polyvinyl alcohol/SiO_2_ for respiration monitoring, as shown in Figure 8a. The respiration was monitored over 60 s for different volunteers and the frequency of SAW decreased during the exhale and increased with the inhale, associated with the respiration process. Moreover, the developed sensor was successfully examined for respiration variation before and after drinking water. Likewise, in [73], the molybdenum disulfide (MoS_2_) based SAW humidity sensor was found to detect the different breath levels, i.e., normal, deep, fast, and rapid. In addition, the same humidity sensor was capable of detecting the noncontact motion of fingers due to the generation of the sharp gradient field of humidity around the fingertip. These outcomes suggest the applicability of SAW humidity sensors in different applications. Moreover, these works indicate that the SAW humidity sensors with high sensitivity, fast response-recovery time, good repeatability, and stability are appropriate for respiratory monitoring and human activity detection.

## 5. Influence of the Sensing Film on the Characteristics of the SAW Device

The overlaid sensing film works as the core element whose existence predominantly characterizes the humidity sensing of the SAW device. The deposition of any kind of sensing membrane permanently changes the device parameters such as resonant frequency, quality factor, signal amplitude, etc. Yan et al. [40] observed a decrease in quality factor and loss in magnitude (S_11_) of about 15.44 dB with the SiO_2_ film coated on the surface of the SAW resonator, as shown in Figure 9a. The insertion loss of the SAW delay line increased from 16 dB to 22 dB with the deposition of PDEB and NaSPF polymer films [50]. Similarly, in [39], the insert loss was found to increase by 8.4 dB with SiO_2_ film coated on the entire SAW device, and the loss was further increased with the deposition of PVA and graphene/PVA on the delay path of the SAW device, as shown in Figure 9b. Moreover, the lower output amplitude and longer delay time were observed for coated SAW devices, as illustrated in Figure 9c.

The decrease in quality factor and loss in magnitude further increases with the absorption of water molecules due to the change in mass loading of sensing film. In [74], with the increase in humidity, a reduction in amplitude and wide bandwidth were observed for GO and MoS_2_/GO-based humidity sensors, as shown in Figure 9d,e. Similarly, Lu et al. [79] obtained the Q factor of the SAW resonator decreased by half (from 1431 to 650) with the deposition of 350 nm thin PVA film on IDC reflectors. Moreover, the peak was further decreased with the increase in humidity and was found to be undetectable for higher %RH, resulting in a small sensing range, up to 78% RH, as shown in Figure 9f. The insertion loss of 60 dB in 10–70%RH is reported for the LiCl-doped TiO_2_-based SAW humidity sensor in [71]. Buvailo et al. [31] compared the magnitude of the SAW response to humidity for PVA and PVP films with different thicknesses. They illustrated that the response drops off at a lower %RH for thicker polymer films, as shown in Figure 9g,h. Thicker PVP film showed high sensitivity with the Insertion loss of 50 dB in the 5–95% RH range. However, some coated sensing materials have not caused a significant change to device characteristics [37,60,61,68,69].

Moreover, the parameters such as thickness, density, viscosity, etc., of the sensing film also influence the characteristics of the SAW device. Mostly the thicker films are utilized to obtain high sensitivity, which increases the mass loading, affecting the device characteristics, i.e., reducing the amplitude of the resonant peak and broader bandwidth [55]. It should not be the case where the sensor exhibits some characteristics of excellent performance while the other parameters do not even suit any practical application.

The significant change in device characteristics is mostly observed for polymer films and thicker coatings. The moderate loss can be acceptable as it originates from the change in mass loading on the SAW device, but the significant loss limits the practical implementation of the sensor. Therefore it is crucial to consider both the sensing characteristics and the performance parameters of the SAW device. There exist few studies that have addressed this issue. In [55] small-sized SAW devices coated with low viscosities, thin PVA films were used to obtain a wide sensing range and narrow bandwidth. Their effort was based on the small mass loading as the thicker and highly viscous films result in large mass loading, broader bandwidth, and decreasing quality factor. Moreover, the reduction in humidity detection range as the peak becomes flat and undetectable. Lu et al. [80] demonstrated that the PVA film coated only on reflectors of SAW resonator presents better device characteristics, including the higher maximum reflection coefficient and narrow bandwidth. However, the sensitivity was low compared to the whole coated SAW resonator due small covering area. Their findings suggested a way to reduce the significant loss of device parameters but at the expense of sensitivity.

## 6. Recommendations for Future Development of SAW Humidity Sensors

As mentioned previously, the deposition of some sensing materials, i.e., polymers, SiO_2,_ etc., causes a significant change in the SAW device characteristics. To the best of our knowledge, the existing humidity sensors are based on traditional SAW device structures. According to [80], the deposition of sensing material only on the reflectors instead of covering the whole area of the SAW resonator results in better device parameters but low sensitivity. The sensitivity is associated with wave energy concentration, which is different at different regions of the SAW device. Powell et al. [81] investigated the mass sensitivity distribution of the 0.6 µm thick photoresist coated on different areas of a two-port SAW resonator consisting of three IDTs. The central IDTs of the device were designed with different periods than the lateral IDTs. They found the greatest sensitivity at the center of the device, which covered only 34% of the active device area and accounted for 84% of the device response. Similarly, Hao et al. [82] analyzed the mass sensitivity variation of SiO_2_ films at different areas of the two port SAW resonator structure consisting of three IDTs as illustrated in Figure 10a. Their results showed that the sensitivity is twice in the central IDTs compared to the lateral IDTs, as shown in Figure 10b–d.

Taking these findings as a foundation, our recommendation for the future development of humidity sensors is to modify the conventional SAW device and deposit the sensing film on the optimal regions of the modified design. This can be carried out by changing the design parameters and configuration of the SAW device so that the wave energy concentrate at the required regions of the device. As discussed before, the reflector-coated SAW sensor provides better device characteristics but low sensitivity, as the wave energy is concentrated toward IDTs. Our idea is to confine a part of the wave at the reflectors by delaying the wave energy and this can be conducted with the variation in the design parameters of the SAW device. So, the deposition of sensing film only on the reflectors can result in improved sensitivity, reduce the mass loading, and also aid in minimizing the significant loss of device characteristics that originate from the large mass loading of sensing film on the entire surface of the SAW device.

## 7. Conclusions

This article summarizes the research on SAW humidity sensors. SAW devices are effectively utilized for humidity detection due to their tremendous features. Current studies are more focused on exploring new sensing materials to get the optimal performance of humidity sensors. Polymers, metal oxides, ceramics, and nanostructured and carbon-based materials are most commonly used as humidity-sensing films for SAW devices. Each of these materials has its advantages and disadvantages. A huge number of polymer-based humidity sensors with good performance characteristics were presented. However, the stability and repeatability of these sensors are adversely affected at high temperatures, which is one of the most sounded shortcomings. Contrarily, metal oxides are found as a good alternative due to their stability in different environmental conditions. Furthermore, ceramics, nanostructures, and carbon-based materials have captured particular attention owing to their porous and large surface areas, which provide the facility of vapor molecules absorption within sensing film, resulting in enhanced sensing characteristics. The overlaid sensing layer is the core element of humidity sensors, as the response highly depends on its interaction with absorbed water molecules. Hence, most of the current studies focus on exploring and altering sensing materials to enhance sensing characteristics such as sensitivity, detection range, linearity, hysteresis, response, and recovery time. As highlighted in this review, the deposition of some sensing materials causes a significant deterioration in the performance parameter of the SAW device, i.e., signal amplitude, bandwidth, insertion loss, Q factor, etc. This loss in device characteristics becomes worst with the absorption of water molecules, especially at high humidity levels, due to large mass loading. Therefore, despite exhibiting good sensing characteristics, such sensors cannot be implemented, as a narrow bandwidth, high-quality factor, and low loss are preferred for practical applications. It is necessary to consider the significance of device characteristics while optimizing the sensing characteristics to make the sensor suitable for practical applications. Few studies addressed this issue and attempted to overcome it by small mass loading effect and deposition of sensing film over the specific regions of the SAW device. Taking these works as a foundation, our recommendation is to modify the traditional SAW device (i.e., altering its design parameters and configuration) and deposit the sensing film over optimum regions of the modified design instead of covering the whole area of the traditional device. This can be a good step in the future development of SAW-based humidity sensors.

## Figures and Tables

**Figure 1 micromachines-14-00945-f001:**
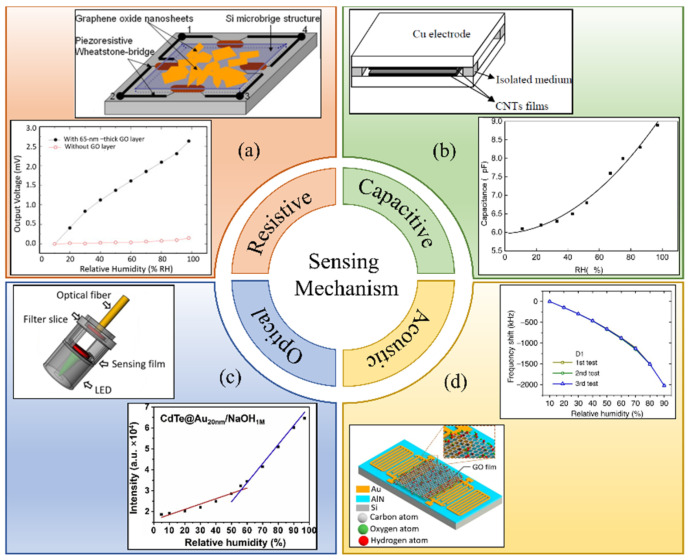
Humidity sensors categories based on sensing mechanism. (**a**) Resistive (reproduce with permission from Ref. [7], Copyright 2012 Elsevier). (**b**) Capacitive (reprinted from Ref. [5]). (**c**) Optical (reproduce with permission from Ref. [10], Copyright 2019 Elsevier). (**d**) Acoustic wave based (reprinted from Ref. [13]).

**Figure 2 micromachines-14-00945-f002:**
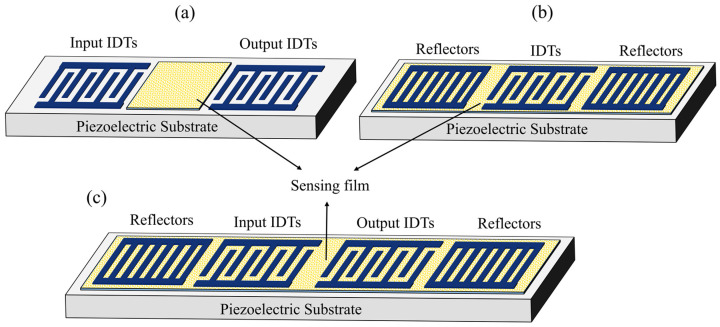
Schematic of SAW humidity sensors based on different configurations. (**a**) Delay line (**b**) One port resonator (**c**) Two port resonator.

**Figure 3 micromachines-14-00945-f003:**
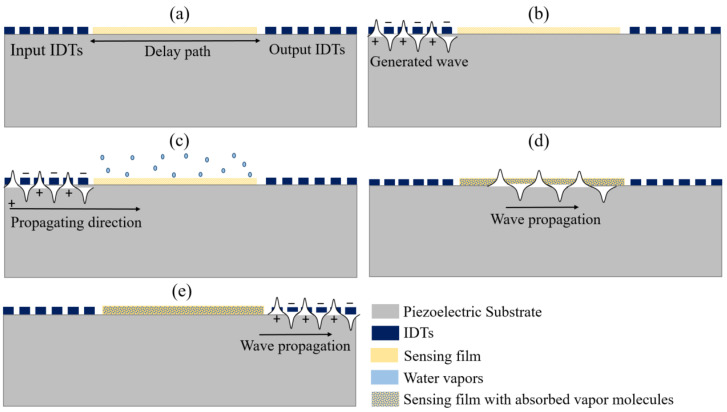
Illustration of the operating mechanism of the SAW humidity sensor. (**a**) Sensing film coated on delay path; (**b**) Application of electrical signal; (**c**) Sensing layer and water molecules Interaction; (**d**) Wave propagation through sensing film with absorbed water molecules; (**e**) Propagating wave at output IDTs.

**Figure 4 micromachines-14-00945-f004:**
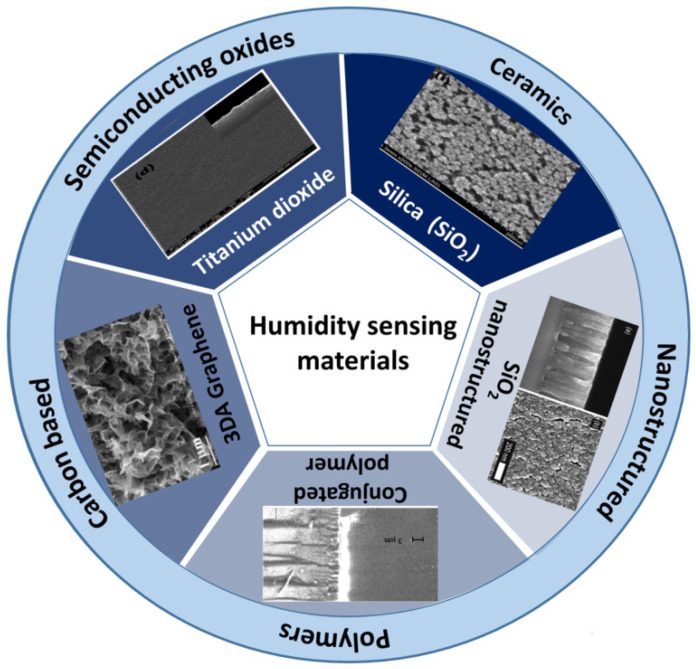
Sensing materials classification for SAW humidity sensors with SEM images. Reproduce with permission from Ref. [38], Copyright 2015 Elsevier, Ref. [46], Copyright 2012 Elsevier, Ref. [47], Copyright 2002 AIP Publishing and Ref. [39], Copyright 2020 Elsevier.

**Figure 5 micromachines-14-00945-f005:**
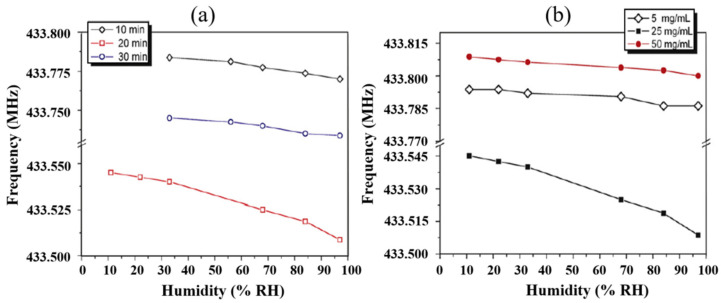
(**a**) The effect of electrospray time (**b**) The effect of concentration of electrospray solution on the frequency response of SAW humidity sensor. Reprinted with permission from Ref. [59] Copyright 2010 Elsevier.

**Figure 6 micromachines-14-00945-f006:**
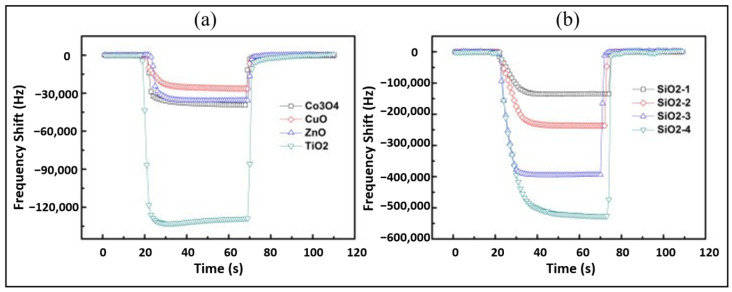
Dynamic responses of SAW humidity sensors. (**a**) Metal oxide films; (**b**) SiO_2_ films. Reprinted with permission from Ref. [38], Copyright 2015 Elsevier.

**Figure 7 micromachines-14-00945-f007:**
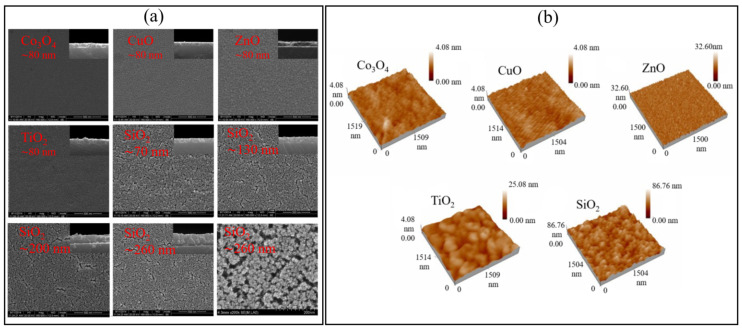
(**a**) SEM images. (**b**) AFM images of Metal oxides and SiO_2_ films. Reprinted with permission from Ref. [38], Copyright 2015 Elsevier.

**Figure 8 micromachines-14-00945-f008:**
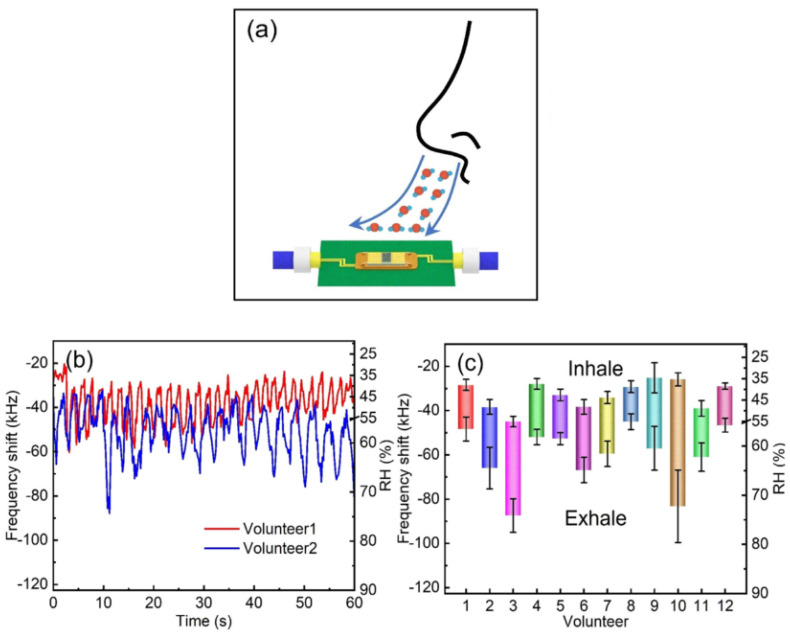
(**a**) Respiration monitoring by utilizing a SAW humidity sensor. (**b**) Real-time breath monitoring of two volunteers. (**c**) Frequency response to humidity for twelve volunteers (reprinted with permission from Ref. [39] Copyright 2020 Elsevier).

**Figure 9 micromachines-14-00945-f009:**
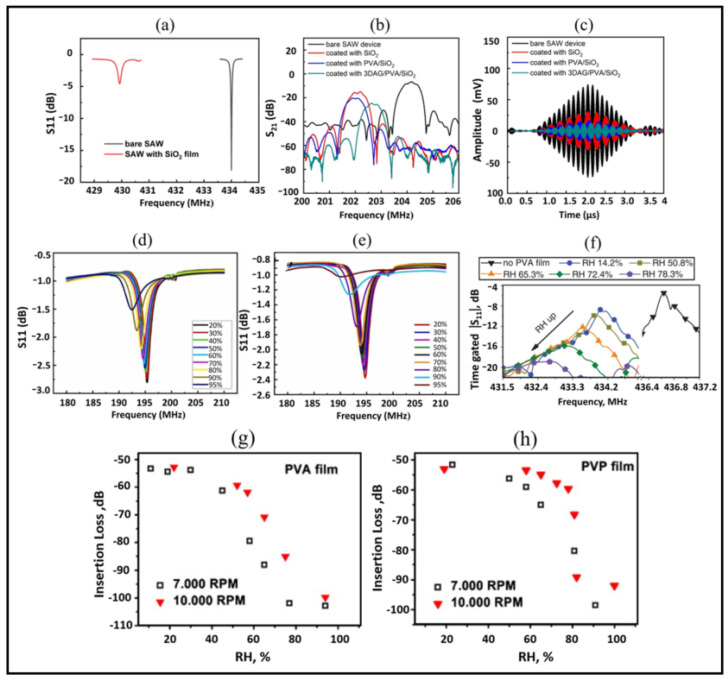
(**a**) S_11_ parameters of the bare and SiO_2_ sensing film-coated SAW resonators (reprinted with permission from Ref. [40] Copyright 2021 John Wiley & Sons). (**b**) S_21_ parameters (**c**) Time response of the bare SAW device and devices coated with SiO_2_, PVA/SiO_2_, and 3DAG/PVA/SiO_2_ (reprinted with permission from Ref. [39], Copyright 2020 Elsevier). (**d**) Humidity response of GO-coated SAW sensor (**e**) Humidity response of MoS_2_/GO-coated SAW sensor (reprinted with permission from Ref. [74], Copyright 2022 Elsevier). (**f**) Humidity response of PVA film coated on IDC reflectors (reprinted with permission from Ref. [79], Copyright 2015 John Wiley & Sons). (**g**,**h**) insertion loss as a function of ambient humidity level of PVA and PVP-coated SAW sensors, respectively (reprinted with permission from Ref. [31], Copyright 2011 Elsevier).

**Figure 10 micromachines-14-00945-f010:**
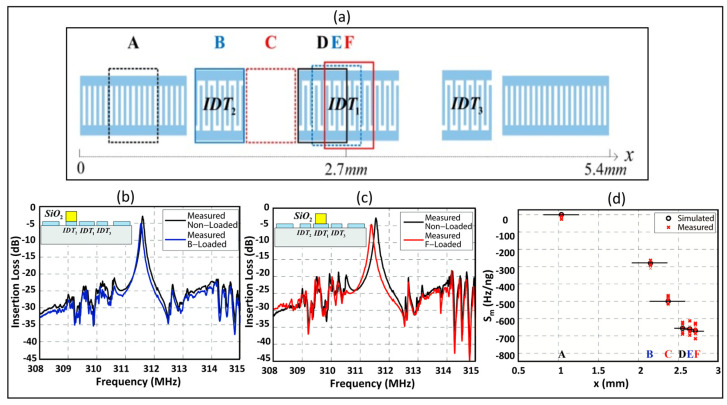
Mass sensitivity variation of SiO_2_ films. (**a**) Two port SAW resonator with three IDTs; (**b**) Measured frequency responses of SiO_2_ deposited on the lateral IDTs; (**c**) Measured frequency responses of SiO_2_ deposited at the central IDTs; (**d**) Mass sensitivity for different areas on SAW resonators, reprinted from Ref. [82].

**Table 1 micromachines-14-00945-t001:** Comparison of different humidity sensor types.

Sensing Technique	Basic Mechanism	Advantages	Disadvantages
Capacitive	Change in the capacitance value due to change in the permittivity of sensing film with the interaction of water vapor molecules	Low power consumption, low cost, simple readout circuit, nonmoving structures, ability to detect the humidity in the entire range, 0–100% range [15]	Temperature dependency, hysteresis issues, and non-linear response
Resistive	Alteration in resistance of sensing membrane when it comes in contact with the water molecules	Simple structure, low cost, and high interchangeability	Temperature dependency, high nonlinearity, poor repeatability, highly sensitive to other contaminants
Optical	Variation in refractive index with the absorption of water molecules	High sensitivity, thermal stability, low attenuation, and chemical inertness [16]	Bulky, high cost [17,18]
SAW	Change in frequency shift due to change in mechanical stiffness and mass of sensing film induced by humidity changes	High sensitivity, high accuracy, low cost, miniaturized, low power consumption	Design challenges associated with temperature sensitivity

**Table 3 micromachines-14-00945-t003:** Sensing characteristics of SAW humidity sensors based on different materials.

Category	Material	Range (%RH)	Total Shift Δf (kHz)	Sensitivity (kHz/%RH)	Response Time (s)	Recovery Time (s)	Hysteresis (%)	**Ref**
Polymer	PDEB	20–85	-	0.4	-	-	-	[50]
	NaSPF	20–85	-	1.1	-	-	-	[50]
	Polyelectrolyte APTS-P	11–97	-	0.4	10	10	-	[59]
	PVP	5–95	-	-	1.5	2.5	-	[31]
	1% PVA	0–99.4	-	4.52 4.97 7.35	30 31 35	40 40 46	0.0003	[55]
	2% PVA	0–99.2					0.001	
	5% PVA	0–98.8					0.003	
	2.5% PVA	60–98	-	-	-	-	1.96	[56]
	5% PVA	10–98					0.26	
	10% PVA	10–95					0.01	
	Fluorinated PI	10–90	332	4.15	7	13	-	[33]
	Nafion	5–95	-	4.5–5 at 0 °C	-	-	±2 at 0 °C	[32]
				6.5–7 at 60 °C			±1 at 0 °C	
Metal Oxide	ZnO	10–90	160	-	-	-	-	[60]
	Ga doped ZnO	10–90	420	-	-	-	-	[61]
	Co_3_O_4_	30–93	50	-	3		-	[38]
	TiO_2_	30–93	125	-	-	-	-	[38]
	CuO	30–93	30	-	50	-		[38]
	ZnO	30–93	40	-	-	-	-	[38]
Ceramics	γ-Al_2_O_3_	3–85	80	0.94	1	3	0.3	[64]
	SiO_2_	30–93	520	-	<10	<10	-	[38]
	Al_2_O_3_	0–95	-	8.67	50	50	0.5	[65]
	SiO_2_	10–80	-	1.14	6	21.3	-	[40]
Nanostructutuerd	PANI nanofiber	5–90	300	-	-	-	-	[41]
	PANI/PVB nanofiber	20–90	-	75	1	2	-	[66]
	MWCNT/Nafion nanofiber	10–80	-	427.6	∼3 s@63%	∼3 s@63%	<1	[67]
	ZnO nanorods	10–90	750	3.5 in 10–50 15.25 in 50–90	-	-	-	[62]
	ZnO nanorods on Ga-doped ZnO seed layer	10–90	970	7.4 in 10–50 16.75 50–90	-	-	-	[63]
	NSOH nanobelts	11–85	2950	-	21	10	-	[68]
	NiO nanoparticles	11–85	5810		23	4	-	[68]
	SnO_2_/MoS_2_ nanocomsposite	10–90	-	0.78	-	-	-	[72]
	MoS_2_ nanomaterial	10–90	-	12.41	0.4	0.8	4.32	[73]
	Bacterial cellulose	10–93	89.8	-	12	5	-	[76]
	MoS_2_/GO composite	20–95	4800	-	6.6	3.5	-	[74]
Carbon-based	GO	20–90	-	-	22 in 20–80	8 in 80–80	<3	[77]
	GO	10–90	-	25.3	<10	<10	<1	[13]
	3DAG/PVA/SiO_2_	5–90	∼140	0.991 in 5–55 2.429 in 55–90	24	14.4	7.8 in 5–85	[39]

## Data Availability

Not applicable.

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
