# Peer review of "Surface Acoustic Wave Humidity Sensor: A Review"

_micromachines, 2023, doi:10.3390/mi14050945_

Round 1

Reviewer 1 Report

This article reviews the sensing materials used for developing surface acoustic wave humidity sensors and their theoretical responses and experimental results. This article also focuses on the influence of overlapping sensing films on the performance parameters of surface acoustic wave devices, such as quality factor, signal amplitude, insertion loss, etc. The authors are advised to consider the following suggestion to further improve the paper quality.

(1) Please provide a comparison of the relative cost and performance of different sensor types to help readers better understand the advantages and disadvantages of different sensor types.

(2) If possible, please provide practical application cases of surface acoustic wave humidity sensors to help readers better understand the practical application effects of these sensors.

(3) Please conduct a more detailed discussion on different coating technologies and deposition conditions, and provide more specific recommended conditions and parameters.

(4) It is recommended to further explore the preparation methods and application scenarios of various materials, and how to optimize and improve their sensitivity, response time, and stability.

(5) It is recommended to translate the research results into practical applications and promote the application of surface acoustic wave sensors and humidity sensors in industry and daily life.

(6) Please explore the influence of different sensing materials on the characteristics of SAW devices, including different coating thickness, viscosity, type, etc.

(7) Research how to reduce changes in mass load and losses while ensuring sensor performance, in order to expand the sensing range and improve sensitivity.

(8) Please further explore the design and manufacturing details of SAW devices in the article, in order to help readers better understand these technologies.

(9) Please further discuss the demand and application prospects of the sensor market in the article, in order to better guide the future development of sensors.

(10) It is recommended to modify the sentences in this article to make readers read more smoothly.

The english writing is fine.

Reviewer 2 Report

This article reviews the development of surface acoustic wave humidity sensors. It presents the operational mechanism, different sensing materials, performance characteristics and their influence on the device’s characteristics. The article is well written; however, the following suggestions are provided to address before publication.

1.       It is suggested to add a table containing the advantages and disadvantages of different sensing materials for better understanding.

2.       Define all the abbreviations in their first appearance. In Tables 1 and 2, some materials are abbreviated while others are not. Please maintain consistency throughout the article.

3.       In Section 5 (recommendations for Future Development of SAW Humidity Sensors), “This can be done by changing by changing the design parameters and configuration of the SAW device so that the wave energy concentrate at the required regions of the device.” Provide some more explanation about how to concentrate wave energy.

4.       If possible, it is recommended to explore the influence of physical parameters of sensing films, such as density, thickness, viscosity, and uniformity, on the characteristics of the SAW device (q-factor, signal amplitude, BW).

5.       In Figure 9, the subtitle is missing for the resonator design structure. 
